# MicroRNA-Based Diagnosis and Therapy

**DOI:** 10.3390/ijms23137167

**Published:** 2022-06-28

**Authors:** Phuong T. B. Ho, Ian M. Clark, Linh T. T. Le

**Affiliations:** 1Biotechnology Department, Ho Chi Minh City Open University, Ho Chi Minh City 70000, Vietnam; bichphuong.hth90@gmail.com; 2Biomedical Research Center, University of East Anglia, Norwich NR4 7TJ, UK; i.clark@uea.ac.uk

**Keywords:** miRNA-based therapy, miRNA-based diagnosis, circulating microRNA, clinical application, prognosis

## Abstract

MicroRNAs (miRNAs) are a group of endogenous non-coding RNAs that regulate gene expression. Alteration in miRNA expression results in changes in the profile of genes involving a range of biological processes, contributing to numerous human disorders. With high stability in human fluids, miRNAs in the circulation are considered as promising biomarkers for diagnosis, as well as prognosis of disease. In addition, the translation of miRNA-based therapy from a research setting to clinical application has huge potential. The aim of the current review is to: (i) discuss how miRNAs traffic intracellularly and extracellularly; (ii) emphasize the role of circulating miRNAs as attractive potential biomarkers for diagnosis and prognosis; (iii) describe how circulating microRNA can be measured, emphasizing technical problems that may influence their relative levels; (iv) highlight some of the circulating miRNA panels available for clinical use; (v) discuss how miRNAs could be utilized as novel therapeutics, and finally (v) update those miRNA-based therapeutics clinical trials that could potentially lead to a breakthrough in the treatment of different human pathologies.

## 1. Introduction

MicroRNAs are small, non-coding RNA molecules that suppress gene expression both by inhibiting protein translation and promoting mRNA cleavage [1]. Since their exploration, a large amount of miRNAs have been described and extraordinary progress has been made toward finding their function as well as their uses in research and clinical practice.

The multi-step process of biogenesis of a miRNA is started in the nucleus and finishes in the cytoplasm (Figure 1). First, the miRNAs are transcribed, mainly by the RNA polymerase II enzymes, and following capped, spliced, and polyadenylated. These steps result in forming a primary miRNA (pri-miRNA) with one or more hairpin structures [2]. Within the nucleus, the pri-miRNA is processed by Drosha (RNAse) and its cofactor DGCR8 into 70- to 100 nt pre-miRNAs which are transported to the cytoplasm by Exportin-5 through the nuclear pores [3,4]. Afterwards, Dicer, another RNAse, cut pre-miRNA to double-stranded RNA in the cytoplasm that embraces the miRNA strand and its complementary sequence. Helicase continuously unwound this duplex miRNA into a single short RNA strand called mature miRNA that is incorporated into the RISC complex containing an Argonaute protein (Ago-2) [5]. This incorporation leads the RISC complex to the 3′UTR of the target mRNA results in mRNA cleavage (if homology is high) or inhibition of translation (normal in mammals) [5]. Besides the above canonical miRNA biogenesis, miRNAs are also processed from the intron of protein-coding gene by the pre-mRNA splicing machinery that is a Drosha/DGCR8 independent pathway [6,7,8]. The expression of these miRNAs, called miRtrons, is correlated with the host gene expression base on their location in either the introns or splice site junctions [9,10,11]. MiRtrons are continuously transported to the cytoplasm and processed by Dicer.

MicroRNAs have been demonstrated to play roles in intercellular communication [12,13,14,15,16]. MicroRNA, which migrate outside the cells and enter body fluids, are called circulating miRNAs [16]. It was shown that approximately 90% of circulating miRNAs form complexes with proteins including Ago2 [12], NPM 1 (nucleophosmin 1) and high-density lipoprotein [13]. The other 10% of the circulating miRNAs are secreted in exosomes [14,17], a form of endosome-derived microvesicles fusing with the plasma membrane [17]. The packaging in either exosome or forming complex with proteins is necessary to protect miRNAs from the digestion of RNases in body fluids [18]. Exosome which contains miRNAs may enter neighboring cells over endocytic uptake, membrane fusion, or integrating with specific receptors in the cell surface and affect mRNA targets remotely from their origin [19,20]. Based on the specific to the cell of origin of exosome content, and the variation depending on the physiological and pathological condition, the circulating miRNAs in exosomes can reflect this [19]. Therefore, the noninvasive diagnosis and prognosis can be developed using circulating miRNAs.

There are several key steps for a circulating miRNA biomarker to enter clinical practice comprising discovery, verification, validation, and clinical application. Though numerous circulating RNA biomarkers had been verified with different populations and disease conditions, they have failed to enter clinic to date. The reasons for this failure can be attributed to the use of different measurement platforms, the normalization strategy used, and other factors such as race, age, sex, sample type, and sample processing method, resulting in considerable inconsistency and irreproducibility between studies. To develop circulating miRNA-based biomarkers for clinical usage, these challenges must be tackled.

MicroRNAs represent a novel treatment approach for numerous human disorders in which the level of a given miRNA in specific tissues could be increased or inhibited. Through derivatization with phosphorothioate linkages and 2’-O-methyl or LNA nucleosides, enzymatic degradation can be delayed, and pharmacokinetics enhanced. However, for efficient application, delivery hurdles still need to be overcome.

This article aims to provide information about the potential for the application of miRNAs in disease diagnosis and therapeutics.

## 2. Intracellular and Extracellular miRNA Trafficking

In order to be secreted outside cells, miRNAs can be loaded into high-density lipoproteins, or packed into exosomes, or bound by Ago2. These complexes can prevent the degradation of miRNAs by RNases and enhance their stability. Interestingly, it was found that the distribution of miRNAs in exosomes is higher than in their origin cells. The mechanisms for sorting of miRNAs into exosomes are largely unknown. Specific sequences in certain miRNAs may instruct their incorporation into exosomes. Four potential mechanisms may be involved. The neutral sphingomyelinase 2 (nSMase2)-dependent pathway was reported to relate to miRNA secretion into exosomes, since gain-of-function of nSMase2 increased the exosomal miRNAs whilst this was reduced with an nSMase2 inhibitor [15,21]. The second pathway is the miRNA motif and sumoylated hnRNPs-dependent pathway, in which GGAG sequence in the 3′ end of the miRNA is recognized by the sumoylated hnRNP family protein and leads to specific miRNAs to be packed into exosomes. This model indicates that the 3′ region of the miRNA contains critical signals for miRNA secretion. A similar mechanism was also reported, e.g., the 3′ miRNA-sequence dependent pathway in which miRNAs with more poly (U) than poly (A) at the 3′ end are preferentially sorted into exosomes. Finally, the miRISC-related pathway has also been reported to have the function in the process of releasing miRNA into exosomes. For loading to HDL, the incorporation of miRNA into HDL might occur extracellularly and involve calcium, cAMP concentrations, and ATP-sensitive potassium channels [22]. However, the exact mechanisms of this loading are unclear.

The function of exosomal miRNAs has also been shown. Whilst they could repress gene expression at distant sites through a mechanism similar to intracellular miRNAs, extracellular miRNAs could also serve as a ligand to activate or suppress the immune system. For instance, miRNA-21 and miR-29a can exert their functions on immune cells via exosomes and further bind and activate TLR8 that results in the activation of NFκB signaling pathway [23]. TLR7 was activated by extracellular let-7 lead to induced neurodegeneration through neuronal TLR7 [24]. Exosomal miR-21, released by cancer cells, triggers TLR8 and NF-κB to activate monoctyes which in turn, release exosomal miR155, which entered into neuroblastoma cells can increase resistance to chemotherapy both in vitro and in vivo [25]. Upon entering the cells, miR-155 suppresses *TERF1,* an inhibitor of telomerase, resulting in increasing telomerase activity and chemoresistance [25,26,27,28].

However, the role of exosomal miRNAs in modulating the immune system is not yet completely elucidated.

## 3. The Potential Application of MicroRNAs in Diagnosis and Prognosis

Circulating microRNAs are being considered as promising biomarkers for many human diseases since they meet several of the criteria for being a preferable biomarker including high specificity, easy accessibility, and sensitivity. MicroRNAs have been found to exist in many biological fluids where they are highly stable [29]. MicroRNAs can easily be extracted from, e.g., blood or other liquid biopsies. MicroRNA also have high specificity for tissue or cell types. For instance, miRNA-122-5p is highly enriched in liver [30] or miRNA-140 is a cartilage-selective miRNA [31]. This feature raises the possibility of using specific miRNAs to determine the initiation and progression of a disease [32,33,34]. The sensitivity of miRNAs has been demonstrated and their levels can vary in accordance with disease progression or with response to therapy. With these advantages, miRNAs give a non-invasive method for accurate diagnosis, prognosis of disease progression, to guide treatment, as well as evaluating responsiveness to treatment.

MicroRNAs have been established as biomarkers for cancer since 2008 in which they were utilized for examination of the diffuse large B-cell lymphoma in patient serum [35,36]. Since then, their putative utility as biomarkers has been proposed for numerous human disorders [37,38,39]. Aberrant levels of circulating miRNAs in human cancers has been reviewed elsewhere [40,41,42,43]. A brief summary of modulated circulating miRNAs in plasma, serum, and blood samples in lung, prostate, breast, colorectal, gastric, and liver cancers were shown in Figure 2.

Numerous studies have identified several circulating miRNAs, which were deregulated in the plasma and serum of cancer patients. For instance, in gastric cancer, miR-20a-5p, and miR-221 were found to increase in both plasma and serum samples. Interestingly, changes of miR-221-3p, miR-378c-3p, and miR-744-5p in serum could aid the prediction of gastric cancer 5 years before any clinical symptoms appeared [44]. In colorectal cancer, a consistent change of miR-21-5p [45,46,47,48], and miR-29-3p [47,49] in patient plasma and serum was reported. Additionally, the level of miRNA 21-5p in the circulation was demonstrated to strongly correlate with that in colorectal cancer tissue [48]. Crucially, the plasma expression of miRNA-21-5p could differentiate colorectal cancer patients with 90% sensitivity and specificity [48]. Moreover, the plasma miR-29a-3p and miRNA-92a-3p levels could potentially provide a good diagnosis with 83% sensitivity and around 85% specificity [50]. In hepatocellular carcinoma, miRNA-122-5p, a hepatocyte-enriched miRNA [30], was identified to consistently change across serum and plasma samples [51,52,53]. The level of this miRNA in serum could help to distinguish hepatocellular carcinoma patients with around 82% sensitivity and 83% specificity [51]. In pancreatic cancer, a few circulating miRNAs displayed potential diagnostic value. The increase in miRNA-196a levels were shown to consistently change in different studies and sample types [54,55]. The overexpression of miR-200a-3p and miR-200b-3p in serum gave sensitivities between 71.7% and 84.4% for detecting pancreatic cancer [56].

Besides miRNAs mentioned above, there are still many miRNAs which change expression in different type of cancers (Figure 2). However, there were a few overlapping miRNAs determined for a given cancer (Figure 2). Therefore, the feasibility of finding a unique circulating miRNA signature for a distinct type of cancer might be difficult. Instead, a panel of circulating miRNAs may be more feasible. It would be necessary to perform large-scale studies on the basis of these panel of miRNAs with well-categorized and diverse populations of patient samples.

The early diagnosis of cancer is critical to improve the end result of therapy and the overall survival rates of patients. Numerous circulating miRNA sets have been reported to have the ability to identify initial stages of cancers with high specificity and sensitivity. For instance, a 2 miRNA set (miR-1254 and miR-574-5p) [57], a t3 miRNA set (miR-197,-182, and -155) [58], a four miRNA set (miR-486-5p, -210, -126, and -21) [59], a 10 miRNA set (miR-320, -223, -222, -221, -199a-5p, -152, -145, -25, -24, and -20a) [60], and an array of 34 miRNAs [61,62] have all been suggested for early detection of non small cell lung cancer. A two microRNA set (miR-378 [63] and miR-199a [64]) and a five-plasma-miRNA signature set (miR-486-5p, -451, -92a, -25, and -16) were potential biomarkers for early detection of gastric cancer; the reduced levels of miR-122 and increased levels of miR-192 in the plasma of gastric cancer patients might be utilized for the early diagnosis of distant metastasis.

Circulating miRNAs could be promising prognosis markers for cancers in which their altered levels can be used to predict the effect of chemotherapy and disease recurrence. In breast cancer, up-regulation of miR-125b can reveal a lower therapeutic response to 5-fluorouracil (5-FU), epirubucin, or cyclophosphamide [32]. Higher levels of miR-210 were indicated to trastuzumab resistance and tumor development [33]. MicroRNA-122 and miR-375 corresponded with the response of neoadjuvant chemotherapy [34]. In prostate cancer, higher levels of miR-21 linked with a resistance to docetaxel [65]. Three elevated miRNAs (miR-141, -146b-3p, and -194) were related to a rapid biochemical recurrence, the increased levels of prostate-specific antigen (PSA) in blood of prostate cancer patients undergoing radiation or surgery. In colorectal cancer, a five serum miRNA signature panel (miR-20a, -130, -145, -216, and -372) were correlated to chemosensitivity/chemoresistance [66]. Higher levels of three miRNAs (miR-27b, -148a, and -326) were related to the non-response of oxaliplatin-based chemotherapy [67]. The miRNA-29c [68] and miR-21 [69] correlated with recurrence in patients. In lung cancer, a four miRNA panel (miR-486, -30d, -1 and -499) was correlated to the overall survival of non small cell lung cancer (NSCLC) patients undergoing surgery or/and adjuvant chemotherapy [70]. The overexpression of miR-125b in serum of NSCLC patients were followed with non-responsive cisplatin-based treatment [71]. Elevated levels of miR-29b and miR-142-3p in the serum of early-stage adenocarcinoma patients vigorously associated with cancer relapse within 24 months [72]. The NSCLC patients with elevated levels of miR-22 in the blood were corresponded with a poor response in pemetrexed-treated therapy [73].

For other human disorders, circulating miRNAs were also reported to be potential markers for diagnosis and prognosis. In Alzheimer’s disease, a 7 plasma circulating miRNA set (miR-545-3p, -301a-3p, -191-5p, -142-3p, -15b-5p, let-7g-5p, and let-7d-5p) could detect Alzheimer’s patients with 95% accuracy from healthy donors [74]. In rheumatoid arthritis (RA), the miR-125b expression in serum and blood of RA patients were increased compared to osteoarthritic and normal controls. High serum miR-125b linked with precious clinical feedback to rituximab treatment [75]. In obesity, elevated levels of circulating miR-222, miR-142-3p, and miR-140-5p and a reduced levels of miR-532-5p, miR-520c-3p, miR-423-5p, miR-221, miR-130b, miR-125b, and miR-15a were observed. These circulating miRNAs were also modulated in obese patients following bariatric surgery-based weight loss therapy [76]. In chronic pain, circulating miR-27b was significantly increased while miR-181a, miR-22, and let-7b were decreased in female patients with migraine [77].

The miRNA-based diagnostic field has much advanced with numerous diagnostic tools already offered to clinicians (Figure 3). These include the miRviewTM Mets panel for detecting of cancers of uncertain or unknown primary origin; RosettaGX Reveal panel to distinguish between indeterminate or benign thyroid nodules; The ThyraMir (miRNA classifier) for identifying of thyroid cancer; CogniMIR is in clinical trial for early diagnosis of Alzheimer’s disease; OsteomiR for checking the hazard of a first fracture in type-2 diabetes and postmenopausal osteoporosis females; and ThrombomiR for assessing platelet function. In the 509 validation sample set, the miRviewTM Mets panel allowed identification accurate to 90% and indicated 88% correlation with the patient’s clinicopathological assessment [78]. The RosettaGX Reveal panel gave a negative predictive value of 91%, a specificity of 72%, and a sensitivity of 85% [79]. The combination of ThyGeNEXT (an oncogene panel for thyroid cancer stratification) and ThyraMir suggests an interesting possible cause of the failing of 15-30% of standard cytological evaluations to differentiate benign from malignant stages [80,81,82]. The ThyraMir gives a positive predictive value of 74%, a negative predictive value of 94%, and a decrease in 85% of unessential surgeries of thyroid cancer. The combination of 19 blood-circulating miRNAs expression in OsteomiR provides a fracture-risk index used for early and follow-up treatment [83,84,85].

## 4. Methodologies and Obstacles in miRNA Measurement

MicroRNA molecules are present as three different types, e.g., the primary, precursor and mature miRNAs. For primary and precursor miRNA, due to their sizes, the detection and quantification of these molecules are similar to mRNAs. The mature miRNA is more difficult to detect due to its small size. With the development of new technology, the detection and quantification of miRNA has become easier and more efficient. Numerous molecular methods have been utilized for detecting and quantifying miRNA including Northern blot, real-time RT-PCR, digital PCR, microRNA microarray, and next generation sequencing. Pros and cons of each method are described in Table 1.

Northern blot can be used to quantify miRNAs and was the first method used to do this. It is an effective method for detecting miRNAs based on miRNA-specific probes and is considered a gold standard for proving whether a new molecule is miRNA because it identifies the size of the RNA. The principle of this method is based on the separation of total miRNA or total RNA itself on a denaturing polyacrylamide gel. After separation, RNA is transferred onto a nitrocellulose or nylon membrane and fixed using heat or UV light. A microRNA specific labelled probe is then hybridized with complementary microRNA on the membrane and detected. The Northern blot implementation process is shown in Figure 4A. Northern blot is widely used in miRNA analysis because it is available in laboratories and does not require special equipment. However, the use of oligonucleotide probes and the large amount of RNA gives this technique poor sensitivity; it is also time consuming. This has been improved by the utilization of locked nucleic acids (LNA) in the probe, increasing the sensitivity and reducing the duration of the experiment ([86,87]).

Quantitative Reverse Transcriptase PCR (qRT-PCR) helps to detect and quantify miRNAs more rapidly and with greater sensitivity as compared to Northern blot. When using qRT-PCR, however, proper design of the primers is crucial when working with distinctive members from a desired miRNA family. For example, miR-20-5p has sequences similar with the miR-17–92 family. The mature miRNA sequence has a size of about 20–25 nucleotides. Therefore, primers for normal PCR, about 20–25 nucleotides in length, cannot elongate when they bind to their target sequences. In order to be able to detect miRNA, several modifications are made in which the length of the miRNAs is increased by adding a poly A sequence or a stem loop. In poly (A) real-time RT-PCR, the length of the mature miRNA is first increased by adding a poly A tail into miRNA 3‘ end through Poly A polymerase. The mature miRNA molecules are then converted into cDNAs by oligodT primers. Quantitative PCR is subsequently performed in which a specific primer for the target miRNA and a universal primer for the poly A are utilized (Figure 4B). In stem–loop qRT-PCR, stem–loop primers are used to perform reverse transcription to convert miRNA to cDNA (Figure 4C). This method could be also combined with the poly (A) method to increase the ability to pair with the target miRNA sequence of the primer stem-loop. For this purpose, the mature miRNA molecule has a poly (A) tail added using the poly (A) polymerase. The miRNA molecule with a poly (A) tail is then converted into cDNA with a stem-loop oligo (T) primer. This method is known as stem-loop poly (A) real-time RT-PCR. The size of cDNA formed by the stem-loop reverse transcriptase or stem-loop poly (A) reverse trascriptase will be around 70–80 nucleotides. With this size, cDNA can be recognized by either SYBR green or Taqman Real-time PCR in which a Taqman probe is normally constructed to bind to the reverse strand of the mature miRNA. Quantitative RT-PCR with LNA primers is similar to the poly (A) real-time RT-PCR. However, this method uses primers in which normal nucleotides are replaced by LNA. LNA is a group of nucleic acid analog that contains a methylene bridge linking the 2′ oxygen and 4′ carbon in the ribose moiety. The use of an LNA primer increases binding affinity and therefore specificity. However, this method gives low amplification efficiency. For miRNAs with low expression levels, their quantification used this method may be not accurate. Quantitative RT-PCR is a powerful tool for miRNA quantification. However, the pre-miRNA or pri-miRNA level are found together with the mature miRNA in total RNA samples. This will reduce the effectiveness of reverse transcription and quantitative PCR. Therefore, small RNAs should be separated from total RNA before performing reverse transcriptase whenever poorly expressed miRNAs are quantified [88].

Digital PCR (dPCR) helps to quantify miRNA by dividing a PCR reaction into thousands of PCR reactions. There are two main dPCR platforms, e.g., droplet-based dPCR (QX100 and RainDrop) and chip-based dPCR (QuantStudio 12k and BioMark) with a determined volume of one nanoliter. The design of primers for digital PCR is similar to the conventional quantitative PCR. Negative and positive results from individual reactions will be statistically performed to determine the target miRNA concentration in initial sample volume. This method is considered to yield absolute quantification since there is no need to build a standard curve as you would in qPCR. Depending on the technology of each company, dPCR has different names. For example, if the separation of individual samples is in small oil droplets, the method is called PCR digital droplet (digital PCR droplet) or dPCR emulsion dPCR. It will be called a chip-dPCR if the reaction solution is split into small holes on a microchip surface. dPCR was shown to give high accuracy and was effective for target miRNA absolute quantification in serum compared to real-time PCR [89].

Simultaneously, a series of miRNA molecules can be detected and quantified using techniques such as microRNA microarray or Next Generation Sequencing (NGS, miRNA-Seq). Microarray and NGS technique are applicable for detecting and screening goal. They can give thorough miRNA expression profiles with acceptable price and high throughput. Microarray can only measure known miRNAs and requires a substantial amount of starting materials. On the other hand, NGS requests less input material, and can identify novel miRNAs. However, library preparation of an NGS sequencing is complex and can add substantial bias to sequencing. Crucially, different library preparation methods can result in variances in output data.

Due to the low concentration of circulating miRNA, qRT-PCR and NGS are the normal platforms applied for miRNA measurement in clinical practice. Different data normalization processes may explicate for several inconsistency and irreproducibility. To address the issue, a proper normalization with reliable internal controls is crucial. The most common normalization approach is added a synthetic miRNA such as cel-miRNA-39. In addition, different studies have used a variety of small RNA transcripts, e.g., miRNA-16-5p, RNU6-1, and 5S, as reference genes but none of these are satisfying or broadly admitted as normalization controls for miRNA measurement. However, the use of unique reference gene is likely deficient for accurately measuring miRNAs and a combination of different control miRNAs is highly recommended.

## 5. MicroRNA Potential Application in Disease Therapy

Expanding knowledge about the role of microRNAs in biology and their dysregulation in many diseases has prompted scientists to investigate their potential use in disease therapy. There were two strategies employed [90]. It could be either miRNA restoration therapy in which a down-regulated or non-functional miRNA would be supplied by a synthetic oligonucleotide or miRNA inhibition therapy where the over-expression of an increased miRNA would be inhibited by antagonists [90].

MicroRNA mimics are double-stranded RNA oligonucleotides exactly copying the mature miRNA duplex in which only the guide strand will target mRNAs whilst the passenger strand will be degraded. However, in some case, strand bias could be introduced into the mimic in which the passenger strand could target mRNAs leading to considerable side-effects [91]. Natural miRNA efficacy is normally limited for therapeutic purpose as they are easily degraded by RNAses and/or they could stimulate the innate immune system through activating Toll-like receptors [92,93]. This problem could be addressed through modifying either RNA phosphodiester or ribose sugar backbones of the synthetic microRNA mimic [94,95]. An appealing site for alteration is the 2′-OH position of the ribose sugar since it is not required for miRNA function [96] and found to be attacked by several nucleases to catalyze RNA degradation [96,97,98]. The most common modifications made to this position include 2′-O-methyl (2′-OMe) [97,99] and locked nucleic acids (LNA) [98]. The RNA phosphodiester backbone would be modified with one to two terminal phosphorothioate linkages [100] (Figure 5A). Besides increasing stability, these changes also help to diminish the innate immune response. Interestingly, to avoid the off-target effect that may be caused by the passenger strand in a miRNA mimic, this strand could be divided into two parts in which each one is too small to well interact with mRNA. Since the level of a miRNA mimic will gradually decrease when transfected into the cell, a miRNA encoding plasmid or virus-based vectors could be exploited for any purpose requiring delivery of an miRNA for a specific period. In addition, to switch on miRNAs at a particular area and in a well-timed manner, caged miRNAs could be used. In caged miRNAs, the two strands of the miRNA mimic are linked with a photocleavable linker and will be idle prior to the photocleavable linker is demolished by UV light illumination.

An anti-miRNA therapeutics aims to arrest the expression of miRNAs in the target tissues. This could be achieved through miRNA inhibitors, miRNA sponges, or small molecular miRNA inhibitors inhibiting of miRNA and mRNA interaction (Figure 5B). MicroRNA inhibitors, which are known as antagomirs [101], miRNA masks [102], and LNA miRNA inhibitors work by specifically binding to the desired miRNA to prohibit its interaction with targeting mRNA. The binding of miRNA inhibitor and their target miRNA to form RNA duplexes precedes to the deterioration of miRNAs by RNAse H. Similar to miRNA mimic, a microRNA inhibitor also requires chemical modifications to increase its stability and reduce immune response. Antagomirs and LNA miRNA inhibitor are both antisense oligonucleotides which are very complementary to the desired miRNA. The LNA miRNA inhibitors have the 2′-O and 4′-C atoms of the ribose ring hooked up through a methylene link, declining the elasticity of the ring and prompting the rigid conformation. These modifications provide nuclease resistance and strengthen binding affinity of anti-miRNAs to their targeting miRNAs [103]. MicroRNA sponges are plasmids with numerous miRNA binding sites [104] by which pairing and eventuating diminishing the endogenous miRNAs. To enhance the lifetime of these sponges, the interacting sites are constructed to be not perfectly paired to the miRNA to prevent the dissection of the sponges by RNAse H (which recognizes DNA/RNA duplexes). Interestingly, the operating mechanism of miRNA sponges is not entirely manufactured as endogenous miRNAs sponges present as circular RNAs. For example, ciRS-7, a circular RNA, has 73 miR-7 binding cites in order to regulate miRNA-7 levels in neurons [105]. The microRNA sponge has been an effective method for determining miRNA functions in vitro. For therapeutic utilizations, however, it becomes difficult because of safety and off-target effects created by exceeding exotic plasmids.

MicroRNA -based therapy is promising. However, there are many challenges associated with miRNA delivery limiting its efficacy. Systemic administration is utilized for drug delivery in pragmatic clinical application. However, miRNA mimics or inhibitors would be degraded by RNAse in the circulation even with chemical modification. Moreover, the miRNA could be taken up by other organs leading to non-specific effects. The dense, extracellular matrix in some tissues could also serve as a physicochemical barrier to prevent cellular entry of miRNA mimics or inhibitors [106]. In addition, the negative charge of nucleic acids and relatively large molecular weight may prevent the passive diffusion of a miRNA or inhibitor through the anionic phospholipid bilayer membrane to enter the cytosol [107]. If it was posted by endocytosis, endosomal liposomal trafficking could result in the microRNA being degraded in the lysosome compartment [107]. Therefore, there is a need for a multifunctional delivery system to overcome all the miRNA delivery hurdles in order to exploit the benefits of miRNA-based therapy.

There are several methodologies for miRNA delivery including conjugation [108,109], virus-associated delivery [109], and nanoparticles [109,110]. Even though the virus-associated miRNA delivery approaches have been experimentally proven to be efficient for cancers, virus-related safety concerned have limited its clinical application for the moment and other non-viral delivery systems seem more promising [109]. The conjugation system in which lipids or cell receptor targeting ligands are directly conjugated to a miRNA, is a good approach for miRNA delivery. Although miRNA-conjugate systems are simple and clearly defined, inadequate dispensation brings about miRNA dominant aggregation in the liver and a high dose requirement for sufficient delivery, limiting their applications. RG-101, an anti-miRNA-122 covalently conjugated with N-acetylgalactosamine was developed for hepatocyte delivery [78]. This drug has entered clinical trial. Nanoparticles have many advantages for miRNA delivery [110]. They typically include a cationic component which complexes with the anionic miRNAs thus protecting them from degradation and approving the interaction with cellular membranes to facilitate cellular uptake [111].

Even though no miRNA-based drug is commercially available, numerous miRNA-based therapies for the treatment of many human diseases have entered clinical trials (Figure 5C, Table 2). These include Miravirsen (or SPC3649, a LNA and phosphorothioate modified antagomiR targeting miRNA-122), RG-101 (a N-acetyl-D-galactosamine-conjugated antagomiR targeting miR-122), RG-125 (or AZD4076, an antagomiR targeting miR-103/107), RGLS5040 (an anti-miR-27), RG-012 (a phosphorothioate, 2′-O-methoxyethoxy modified antagomiR targeting miR-21), MRG-201 (a LNA miRNA-29b mimic), MRX34 (a miRNA-34a mimic), MRG-106 (a LNA antagomiR targeting miR-155), MRG-110 (a LNA antagomiR targeting miR-92), MesomiR (a miR-16 mimic), ABX464 (a small molecular compound triggering miR-124 expression).

MicroRNA 122 could enhance hepatitis C virus transcription by binding to its two binding sites in the 5′-UTR of virus genome [112,113]. Miravirsen and RG-101 (antagomiRs targeting miR-122) showed good efficacy in which patients receiving these drugs had undetectable HCV RNA levels in their bloods [78,114]. Miravirsen has gone through phase 2 clinical trial whilst RG-101 is on hold due to several severe side effects [114,115].

RG-125 (also known as AZD4076), RGLS5040 and RG-012 are antagomiRs targeting miR-103/107, miR-27, and miR-21 [116,117,118], respectively. RG-125 and RGLS5040 aim to treat of nonalcoholic steatohepatitis, and cholestatic diseases, respectively, whilst RG-012 is for the fibrogenesis of organs associated with Alport syndrome [116,119]. However, the development of these three was suspended.

The miR-29 family (miR-29a/b/c) is reported to decrease expression in fibrotic diseases in which the miRNA inhibits the buildup of the extracellular matrix [120,121,122]. Interestingly, the miR-29 level could be restored by MRG-201 (also known as Remlarsen), a LNA RNA mimic delivered by intradermal injection [123]. Similar to the miR-29 family, the miR-34 family (miR-34a/b/c) level could be rescued in several cancers, e.g., renal cell carcinoma, acral melanoma and hepatocellular carcinoma by MRX34, a double stranded RNA encapsulated into a liposome-formulated nanoparticle [124].

MRG-106 (Cobomarsen, a LNA antagomiRs) and MRG-107 (an antagonist) both target miR-155. Whilst MRG-106 aims for treating certain types of lymphoma and leukemia [125], MRG-107 is for alleviating symptoms and extending survival associated with amyotrophic lateral sclerosis [117,126]. However, only MRG-106 is in phase 2 while MRG-107 has not yet entered clinical trials [119,127,128]. MRG-110 (a mixer of LNA and DNA antagomiR entirely altered with phosphorothioate internucleotide bridges) target miR-92 in order to cure ischemic conditions such as heart failure [123].

Mesomir, is a miRNA mimic for substituting miR-16 which is repressed in various cancers such as malignant pleural mesothelioma [129]. It has completed its phase 1 clinical trial and is following phase 2 [127]. ABX464 is a small molecular compound prompting the overexpression of miR-124 to reduce the feature of inflammatory colon in patients resisting to corticosteroids and anti-TNF biologics [128,130]. It is in phase 2a and 2b clinical trial for Crohn’s disease and ulcerative colitis, respectively [131,132].

## 6. Conclusions

The exploration of miRNAs has initiated a new perspective on regulating gene expression after transcription. Circulating miRNAs have tremendous prospective to be used as biomarkers for diagnosis and prognosis. However, to establish their reliability, numerous issues need to be addressed. First, the role of circulating miRNAs in the pathology of diseases warrants thorough study. Second, lack of standardized procedures, for, e.g., RNA isolation, sample preparations, and reference controls, leads to difficult for comparing data between studies, thus it is necessary to establish the reference procedures for measuring circulating miRNAs. Third, detailed understanding of miRNA biosynthesis and extracellular trafficking pathways is urgent to understand the origin of miRNAs in healthy individuals and patients as well as to evaluate the correlations between circulating miRNAs and established markers for disease early detection.

Although the potential of miRNA-based therapy is fascinating, much more needs to be addressed. Investigation of microRNA target genes and its functions are crucial for the outline of miRNA-based treatments in different human pathologies. The development of miRNA mimetics and miRNA inhibitors is a good selection for either functional recovery or antagonization of endogenous miRNAs. However, high doses of these exotic miRNA mimics and inhibitors could switch on the innate immune response, resulting in increased expression of numerous cytokines. In addition, many modifications are under investigated to increase stability of miRNA mimics or inhibitors during delivery. These problems need to be resolved for effective future application of miRNA-based therapies.

## Figures and Tables

**Figure 1 ijms-23-07167-f001:**
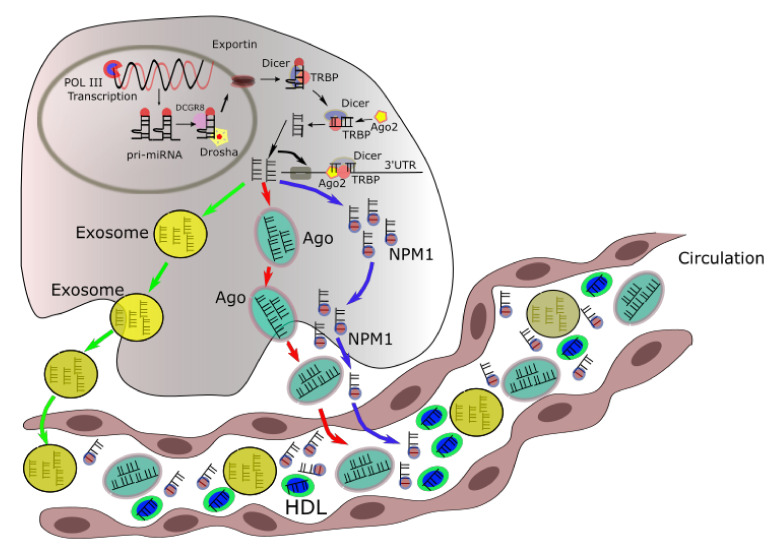
miRNA biogenesis and extracellular export. MicroRNAs are first transcribed by RNA polymerase II as primary (pri-miRNA) transcripts in the nucleus. The pri-miRNA are cut by the Drosha/DGCR8 complex to pre-miRNA precursors that transport into the cytoplasm by Exportin. In the cytoplasm, pre-miRNAs are further cut by Dicer in a complex with TRB, resulting the miRNA duplex. The mature miRNA incorporates with an AGO protein forming miRISC complex which will interact with the target mRNA and suppress its expression. Three ways that miRNAs can be undergone into circulation: mediated by extracellular vesicles (exosomes) (green arrows) or other RNA-binding proteins such as Ago (red arrows) or NPM1 (blue arrows). HDL, high-density lipoprotein. NPM1, nucleophosmin 1. POL III, RNA polymerase. TRB, T cell receptor beta.

**Figure 2 ijms-23-07167-f002:**
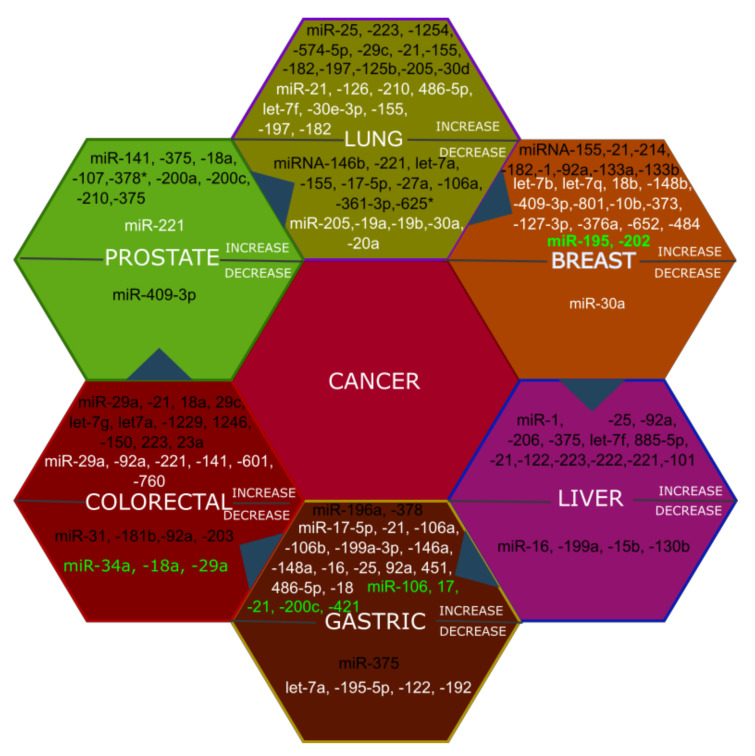
Summary of modulated circulating microRNAs in several prominent cancer types including breast cancer, lung cancer, prostate cancer, colorectal cancer, liver cancer. Circulating miRNAs were from serum (black), plasma (white), and blood (green) of both cancers diagnosed and control counterparts with their levels (either increase or decrease) shown.

**Figure 3 ijms-23-07167-f003:**
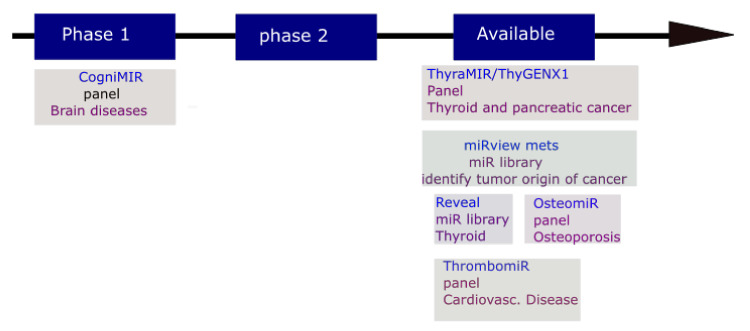
miRNA-based diagnostic tools available in clinic.

**Figure 4 ijms-23-07167-f004:**
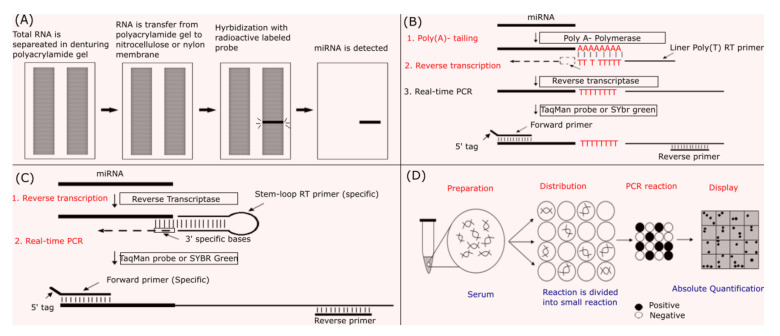
Biological molecular methods for microRNA detection and quantification. (**A**) Northern blot; (**B**) poly (**A**) real-time reverse transcriptase PCR; (**C**) stem–loop real-time reverse transcriptase PCR; (**D**) digital PCR.

**Figure 5 ijms-23-07167-f005:**
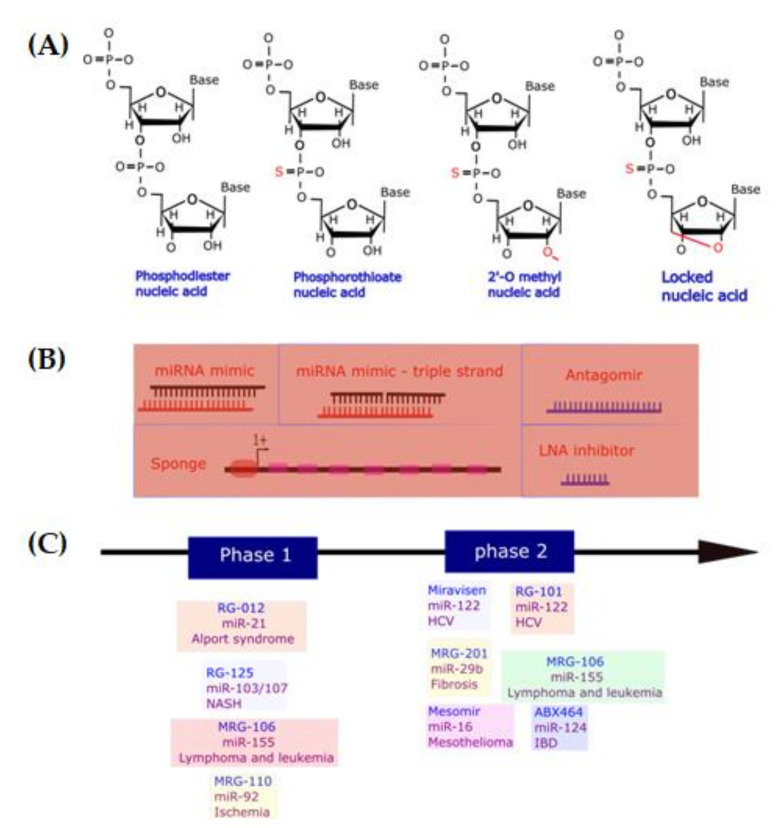
(**A**) Chemical modifications of oligonucleotides; (**B**) miRNA replacement therapy using miRNA mimics, miRNA mimic triple strand; miRNA inhibition therapy using miRNA antagomirs, miRNA inhibitors, and miRNA sponges; (**C**) Current miRNA-based therapy.

**Table 1 ijms-23-07167-t001:** Pros and cons of different methodologies utilized for circulating miRNA quantification.

Methodologies	Pros	Cons
Northern blot	Could determine size of the miRNAs The gold standard for novel miRNA confirmation	Require large amount of RNA Poor sensitivity Time consuming procedure Could only detect a single miRNA at a time Quantification is challenge.
Quantitative Reverse Transcriptase PCR	Fast and great sensitivity Require small amount of RNA	Could not determine size of miRNAs Properly design primers and probe could be challenge Relative quantification Could only detect a single miRNA at a time
Digital PCR	Fast and great sensitivity Absolute quantification Require small amount of RNA	Could not determine size of miRNAs Properly design primers and probe could be challenge Chips and reagents are expensive Detect and quantify a given miRNA at the same time
Microarray	Detect and quantify a series of miRNAs at the same time	Require large amount of RNA Only measure know miRNAs
Next generation sequencing	Detect and quantify a series of miRNAs at the same time Require a reasonable amount of RNA Could identify novel miRNAs	Complex technique Library preparation is complex and can introduce substantial bias to sequencing

**Table 2 ijms-23-07167-t002:** MicroRNA based therapies for the treatment of human diseases and their stages of development.

Therapeutic Names	Treatment for Human Disease	Target miRNAs	Stage of Development
Miravirsen	HCV infection	AntagomiRs targeting miR-122	Phase II
RG-101	HCV infection	AntagomiRs targeting miR-122	On hold
RG-125 (AZD4076)	Nonalcoholic steatohepatitis	AntagomiRs targeting miR-103/107	On hold
RGLS5040	Cholestatic diseases	AntagomiRs targeting miR-27	On hold
RG-012	Fibrogenesis of organs associated with Alport syndrome	AntagomiRs targeting miR-21	On hold
MRG-201 (Remlarsen)	Fibrotic diseases	LNA RNA mimic miR-29a	Phase I
MRX34	Cancers	RNA mimic miR-34a	Phase I
MRG-106 (Cobomarsen)	Lymphoma and leukemia	LNA antagomiRs targeting miR-155	Phase II
MRG-107	Amyotrophic lateral sclerosis	AntagomiRs targeting miR-155	Preclinical
MRG-110	Ischemic conditions such as heart failure	Mixer of LNA and DNA antagomiR targeting miR-92	Phase I
Mesomir	Malignant pleural mesothelioma	RNA mimic miR-16	Phase II
ABX464	Crohn’s disease and ulcerative colitis	Compound inducing miR-124	Phase II

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
