# Peer review of "MicroRNA-Based Diagnosis and Therapy"

_ijms, 2022, doi:10.3390/ijms23137167_

Round 1
Reviewer 1 Report
How does miRNA bind to high-density lipoprotein? Does miRNA also bind to other lipoproteins such as LDL?
How does miR155 increase resistance to chemotherapy?
Why is miRNA highly stable in many biological fluids? Is it resistant against RNase? Authors should add the mechanism.
Are most of the increased miRNAs in serum of cancer derived from cancer cells? Why are some of miRNA in cancer decreased?
Why are specific miRNAs associated with resistance to therapies?
Reviewer 2 Report
This manuscript, written by Dr. Ho PTB et al., review type, with the title of “MicroRNA–based diagnosis and therapy” makes a thorough description of the potential use of microRNAs in a clinical environment. The manuscript is well written, has several tables and figures, and contains enough references.
This manuscript focuses on RNA interference. RNAi is an evolutionarily conserved mechanism in bacteria and eukaryotes that reduces gene expression in response to small (approximately 20 to 25 nucleotides) exogenously or endogenously derived double-stranded RNAs (dsRNAs). The sequence-specific silencing of genes by RNAi can be triggered by different types of RNAs, including siRNAs, short hairpin RNAs (shRNAs), microRNAs (miRNAs), and other noncoding RNAs (ncRNAs) such as long ncRNAs and pyknons (repeated elements frequently found in the 3' untranslated regions of genes).
Two major concerns with RNAi therapies are their short half-life (due to rapid degradation) and their potential for immune stimulation:
(a) Many modifications are under investigation for increasing the stability of siRNA constructs, including chemical modifications of the RNA bases or backbone, conjugation to nanoparticles or lipids, or use of a viral vector to deliver the sequence for an shRNA.
(b) High doses of siRNAs switch on the innate immune response and the production of cytokines.
Comment 1: I think that these two major concerns could be added to the conclusions of the manuscript.
There are known applications of gene silencing: hemophilia, hemoglobinopathies, huntington disease, ALS, amyloidosis, and Friedreich ataxia.
Comment 2: Could you please confirm that these diseases can be mentioned in this review?
Comment 3: In Figure 2 and the associated text, the authors mention circulating microRNAs. The title of this section is “The potential application of micrornas in diagnosis and prognosis.” Could you please confirm that all information in this section is about “circulating” micro RNAs. As I understand, there are also many clinicopathological correlations between microRNAs in different types of cancer from diagnostic tissue biopsies. Should that information also be included in this review manuscript?
Comment 4: Page 4 of 19. Lines 119–120. Regarding the sentence “MicroRNAs have been reported as biomarkers for cancer since 2008 in which they were utilized for investigating of the movement large B-cell lymphoma in patient serum [32, 33].” I am sorry, but I do not fully understand the meaning of this sentence (movement?).
Comment 5: Section 5 is very dense (in my opinion). Could you please summarize it with a table?
Comment 6: A Table of the interventional clinical trials for microRNAs could be added.
Comment 7: If appropriate, you may add information about miRNAs and covid-19 treatment. https://doi.org/10.1002/cbin.11653
Round 2
Reviewer 1 Report
The manuscript was appropriately modified, and there is not more comment.